# Infrared Cirrus Detection Using Non-Convex Rank Surrogates for Spatial-Temporal Tensor

**Shengyuan Xiao [1,2], Zhenming Peng [1,2,*] and Fusong Li [3]**

1   School of Information and Communication Engineering, University of Electronic Science and Technology of China, Chengdu 611731, China

2   Laboratory of Imaging Detection and Intelligent Perception, University of Electronic Science and Technology of China, Chengdu 611731, China

3   Xi'an Institute of Electromechanical Information Technology, Xi'an 710065, China

*   Correspondence: zmpeng@uestc.edu.cn; Tel.: +86-02883208185

**Abstract:** Infrared small target detection (ISTD) plays a significant role in earth observation infrared systems. However, some high reflection areas have a grayscale similar to the target, which will cause a false alarm in the earth observation infrared system. For the sake of raising the detection accuracy, we proposed a cirrus detection measure based on low-rank sparse decomposition as a supplementary method. To better detect cirrus that may be sparsely insufficient in a single frame image, the method treats the cirrus sequence image with time continuity as a tensor, then uses the visual saliency of the image to divide the image into a cirrus region and a cirrus-free region. Considering that the classical tensor rank surrogate cannot approximate the tensor rank very well, we used a non-convex tensor rank surrogate based on the Laplace function for the spatial-temporal tensor (Lap-NRSSTT) to surrogate the tensor rank. In an effort to compute the proposed model, we used a high-efficiency optimization approach on the basis of alternating the direction method of multipliers (ADMM). Finally, final detection results were obtained by the reconstructed cirrus images with a set threshold segmentation. Results indicate that the proposed scheme achieves better detection capabilities and higher accuracy than other measures based on optimization in some complex scenarios.

**Keywords:** cirrus detection; tensor robust principal component analysis; spatial-temporal tensor; non-convex tensor rank surrogate; infrared imagery

## 1. Introduction

The earth observation infrared system is a significant component of the remote sensing application. It significantly influences infrared guidance, remote sensing, missile warning, etc. [1–4]. The infrared detector has the advantages of strong adaptability, good portability, small size, and ease of concealment [5]. As the infrared imaging detection system has been significantly improving, some scholars have developed new target detection recognition algorithms [6–13]. However, some areas or scenes in nature also generate high levels of radiation, which means that they may show similar characteristics to a real small target. This will lead to early warning system false positives, which will interfere with small target detection. Because of the sun's radiation, cirrus will produce a large amount of radiation, which will be imaged in the infrared image together with the real small target. To reduce the false alarm rate, it is essential to research cirrus features and detection methods.

The traditional methods of cloud detection are separated into three classes: pixel-level threshold, texture analysis and the statistical-based method [14–17]. These methods use space, time and frequency domain information, or calculate one or more appropriate thresholds based on a physical model, wavelength difference, etc. According to the obtained thresholds, the cirrus is distinguished from other parts of the image [18–21]. Because of the infrared radiation characteristics of cirrus, almost all of these traditional methods use infrared band images to detect the cirrus.

Besides using spectral or physical properties, since remote sensing technology, computer vision and artificial intelligence are undergoing a spurt of progress, some scholars have developed many new detection methods for cirrus detection [22–27]. Machine learning has improved the detection accuracy within a certain range; however, since the real cirrus scene lacks sufficient data to train the model, the method based on machine learning cannot achieve great detection performance in real scenes. Moreover, some scholars also introduced robust principal component analysis (RPCA) to cirrus cloud detection, but due to the design of the models, these studies have not yet achieved satisfactory results, especially in some complex scenarios [28,29].

Recently, optimization-based methods for target detection such as sparse representation (SR) have gradually been favored by scholars and successfully used in infrared imagery [30–35]. Low-rank sparse decomposition of a matrix or tensor involves regarding the infrared image as a linear superposition of two different parts: background component and target component. This type of method focuses more on the infrared image feature and utilizes the properties of the background and target. The background component has non-local self-similarity, which can be treated as a low-rank component. The proportion of the target component is small and can be treated as a sparse component. Using the above theory, the detection problem is converted into a matrix decomposition problem. For infrared images containing cirrus, the cirrus are still sparse compared to the background component. The background component is low-rank, so a cirrus detection problem can be converted into a matrix decomposition problem.

A traditional small target detection measure using RPCA and SR is often used for single frame image detection. Due to the small dimension and weak pixel levels of infrared small targets, the sparseness of infrared small targets is high in single frame images. However, due to the different imaging forms of cirrus, there are large cirrus. In a single frame image, these forms of cirrus may have poor sparsity. Since the background information of the infrared image in the same region at different times does not change much, and the cirrus may have morphological changes due to time process and displacement due to motion, adding time information can effectively improve the discrimination between low-rank components and sparse components in an established measure.

Considering an introduction of sequence images for detection, the matrix-based low-rank sparse method cannot meet the processing method of multiple images. Therefore, the theory of tensor recovery is considered.

The tensor, as a high-dimensional form of the matrix, is able to be directly utilized in high-dimensional data. Scholars have successfully used tensors to process color images, video and hyperspectral images [36–42]. Tensors are faster and more efficient at solving optimization problems when compared to matrices. By introducing a tensor into RPCA, scholars have proposed a tensor-robust principal component analysis model (TRPCA) [43]. Sequence images or videos with a temporal order can be treated as a three-dimensional tensor, so the TRPCA model can be applied in infrared imagery.

One of the difficulties of TRPCA is the representation of the tensor rank. Considering the non-local self-similarity in an infrared image background, the tensor composed of the background component has a low rank [44]. Tensor rank is the most direct means of measuring low-rank characteristics. However, the definition of matrix rank cannot be directly extended to tensors, and no direct definition can be used for the tensor rank.

Another difficulty is that most of the tensor rank calculation problems are NP-hard. For example: CP decomposition cannot be directly calculated. Therefore, scholars consider using tensor-rank convex relaxation or nonconvex tensor-rank substitution to represent the rank of tensors. Huang [45] used the sum of nuclear norms (SNN) in tensor rank but could not achieve a great effect in a complex background. Lu used tensor nuclear norm (TNN) in tensor rank [46], but each singular value is endowed with the same weight. However, big singular values include the main information of the image, while small singular values are caused by noise. Both SNN and TNN are convex relaxation-of-rank, which limits their performance. Recently, tensor rank non-convex surrogates (NRSs) based on the Laplace

function have been proposed and successfully used [47]. This non-convex surrogate based on the Laplace function can help every singular value obtain a proper weight according to its values, which can better represent the rank of the tensor. As shown in Figure 1, Laplace functions can better approximate the $l_0$ norm than the $l_1$ norm. Guan [48] applied the non-convex surrogate based on the Laplace function to the infrared target detection. Inspired by his strategy, we propose a measure for cirrus detection based on visual saliency and non-convex spatial-temporal tensor rank surrogate (Lap-NRSSTT). The proposed method can use information embedded in a spatial-temporal structure and multiple regularization parameters to obtain better performance. The primary contributions in the paper could be summarized as bellow:

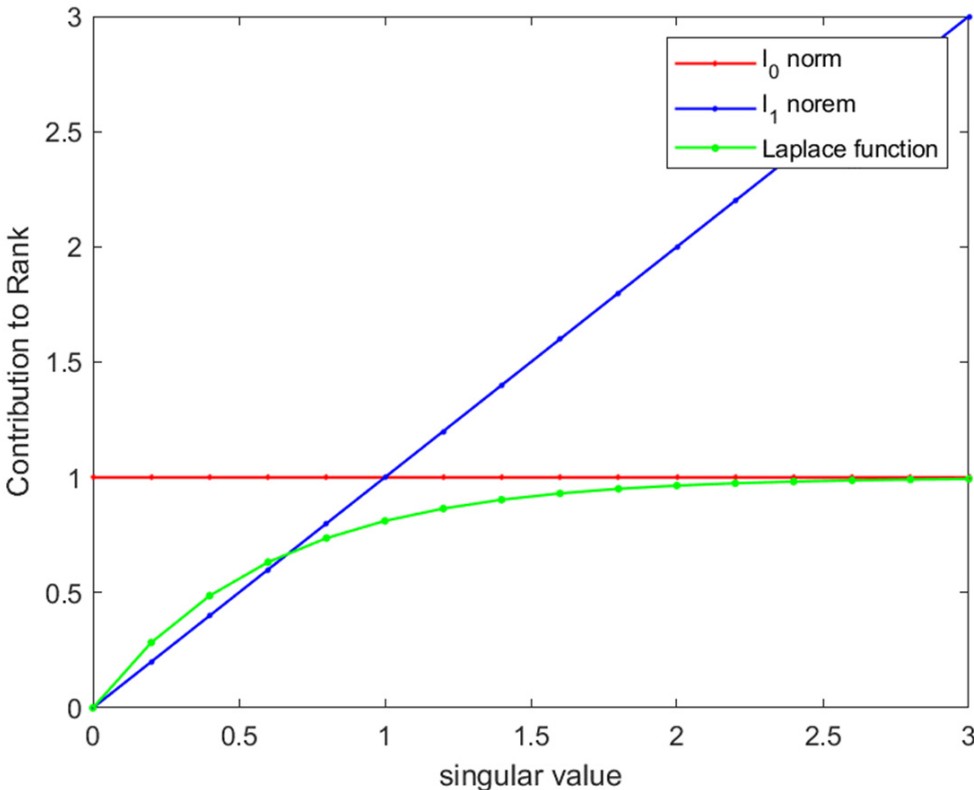

**Figure 1.** Comparison of the contribution of the $l_0 - norm$, $l_1 - norm$, and the Laplace function to the rank.

1. Considering the infrared imaging characteristics of cirrus scenes, a spatial-temporal tensor (STT) model was built, so that a low-rank sparse decomposition method could be effectively used in an infrared cirrus detection scheme;
2. To obtain an easy-to-calculate tensor rank, the NRSs using the Laplace function are applied to the STT (Lap-NRSSTT) completion for infrared imagery; It preserves the details of the cirrus and suppresses noise with smaller singular values;
3. To reduce the time complexity, a mask based on visual saliency is constructed, so that the optimization-based scheme can quickly reach the convergence stop condition with great detection performance.

The rest of this paper is organized as follows: In Section 2, details of the proposed model are proposed, the optimization of the model is designed, and the whole detection process for infrared sequence images is given. Section 3 proves the validity of the proposed measure through some single variable measurement and comparative experiments. In Sections 4 and 5, we summarize the paper and discuss our future work.

## 2. Materials and Methods

In this section, we presented the Lap-NRSSTT model for infrared cirrus detection in infrared imagery.

### 2.1. Construction of STT Model

An infrared cirrus image could be divided into these parts: the infrared image matrix $D$, the background matrix $A$, the cirrus matrix $S$ and the noise image matrix $N$. The definition of the model is as bellow:

$$D = A + S + N \tag{1}$$

To introduce TRPCA into cirrus detection, it was necessary to construct an STT model for detection. Under the tensor model, the infrared cirrus image model could be expressed as bellow:

$$\mathcal{D} = \mathcal{A} + \mathcal{S} + \mathcal{N} \tag{2}$$

where $\mathcal{D}, \mathcal{A}, \mathcal{S}, \mathcal{N} \in \mathrm{R}^{m \times n \times k}$ are the input tensor, background tensor, cirrus tensor and random noise tensor, respectively. The variables m, n and k represent the size of the tensor.

When constructing the tensor model, the infrared small target algorithm would traverse the whole image using a window of size $m \times n$. The obtained image patches would be arranged to form a tensor of size $m \times n \times k$. To better explain the subsequent steps, we explained some of the patch-tensor model construction steps: Considering that the experimental image we used is often preprocessed into an image of equal length and width, a sliding window of $m \times m$ was used, and to make sure that the obtained image patches did not have overlapping parts, the sliding step was also set to m. The whole image was traversed by a sliding window. By using this method, we could reduce redundant information to a certain extent. In this way, each frame of the image would obtain $Num$ image patches.

When used for cirrus detection, due to the different imaging forms of cirrus, there were large cirruses. The sparsity may be poor in a single sliding window. Because the background information of the infrared sequence image did not change much, the sparsity could be improved according to the change of the cirrus in the same position at a different time. Therefore, we considered constructing a tensor model with spatial-temporal characteristics.

To fully utilize the relevance of moving targets in the time domain, the order of image patches would be carefully considered. Here we consider three schemes: $i$ and $j$ were defined as follows: $i$ was the frame index and $j$ was the image patch index. The first scheme was to arrange all the image patches of one frame from small to large according to the patch index $j$, then continue the arrangement of the image patches of the next frame. This measure did not fully utilize the similarity of image patches at the same position between different frames. The second method combined the $j$-th image patch of each frame into a tensor $\mathcal{M}$ of size $m \times m \times i$ in the order of frames, and then arranged $Num$ tensors $\mathcal{M}$ to obtain the final tensor $\mathcal{M}_{Num}$ of size $m \times m \times a$ ($a = i \times Num$) according to the order of the image patch index $j$ from small to large. However, the displacement result from cirrus motion is often reflected on the surrounding patches of the patches with the same image patch index $j$ in adjacent frames. If the tensor was constructed according to the above method, there would be a lot of image patches between $D_i^j$ and $D_{i+1}^{j+1}$, which destroyed the local correlation of each frame in infrared images. Combining the advantages and disadvantages of the above measures, the STT was constructed by the method shown in Figure 2.

Firstly, the spatial patch tensor was constructed with the image patch $D_i^j$ and its surrounding patches. The blue image patches represented the image patches of the current frame $D_i$. The corresponding construction process is shown in Figure 3.

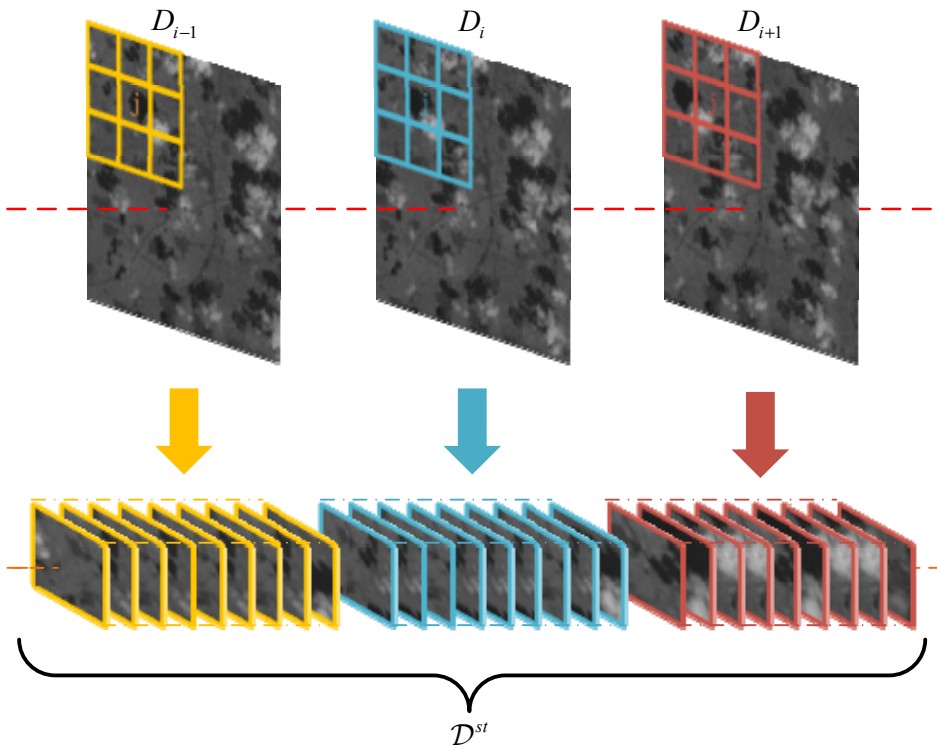

**Figure 2.** Spatial-temporal tensor construction.

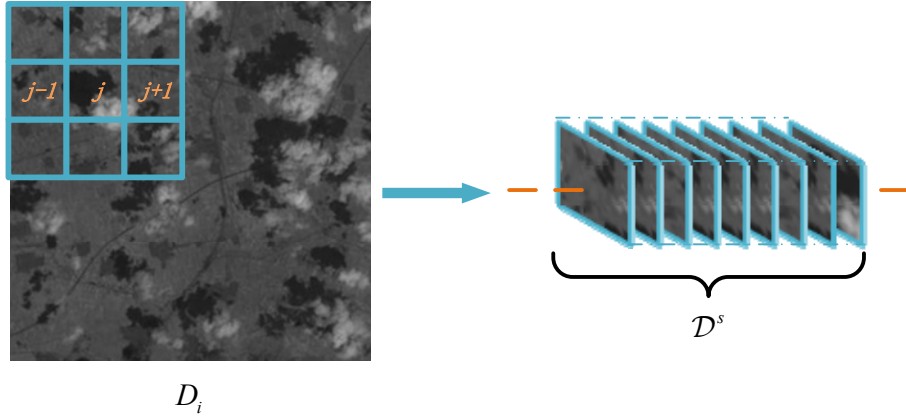

**Figure 3.** Spatial patch-tensor construction.

Then, the time patch tensor was constructed according to the following scheme: After selecting $2 \times s$ adjacent frames $D_{i+y}$ of the current frame, where $y \in [-s, 1] \cup [1, s]$, every frame $D_{i+y}$ was used to build a group of tensors according to Figure 3, labeled as $\mathcal{D}^{st}_{i+y}$. The temporal patch tensor was constructed by $\mathcal{D}^{st}_{i+y}$ and arranged according to the frame index.

STT $\mathcal{D}^{st}$ could be shown as:

$$\mathcal{D}^{st} = \mathcal{A}^{st} + \mathcal{S}^{st} + \mathcal{N}^{st} \tag{3}$$

Thus, we established a new STT model, and fully utilized the spatial-temporal information. We would then analyze the properties of the divided parts in infrared imagery.

Background patch tensor $\mathcal{A}^{st}$: For a three-dimensional tensor, we could obtain the expansion matrix of its various modes and calculated all singular values of the expansion matrix. When singular values rapidly descend to near zero rapidly, it means that the expansion matrix has the characteristics of low rank. We unfolded STT $\mathcal{D}^{st}$ and calculated the singular values of three mode, as shown in Figure 4. There was no doubt that the curve

in the image would rapidly decrease to zero, which meant that our proposed tensor model has low rank characteristics in its multiple modes. Therefore, we could make a hypothesis on the background as follows:

$$\text{rank}(\mathcal{A}^{st}) \leq a \tag{4}$$

where $a$ was a low-rank constant that constrains the background tensor. Generally, $a$ was larger in a complex background.

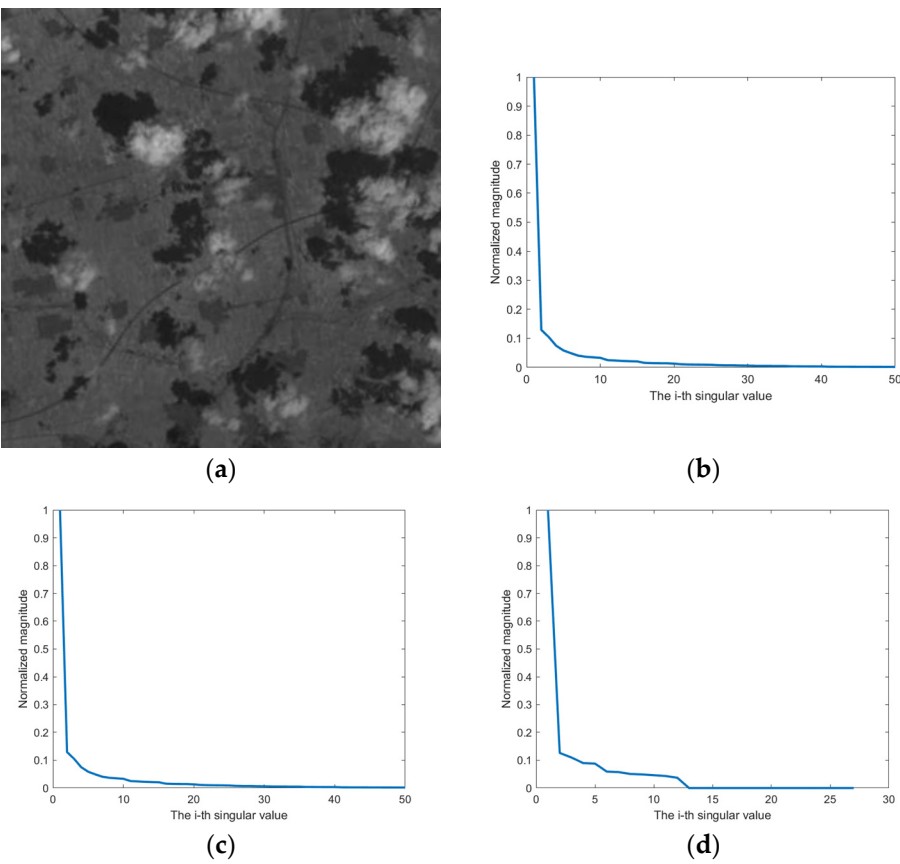

**Figure 4.** The i-th singular values of unfolding matrices for infrared tensor $\mathcal{D}^{st}$. (**a**) Cirrus image. (**b**–**d**) Singular values of mode-1, mode-2, and mode-3 unfolding matrices.

Cirrus patch tensor $\mathcal{S}^{st}$: The distribution range of the cirrus on the image was relatively small compared with the background; in the infrared video frame, the position shift or the morphological change of the cirrus caused by the movement of the cirrus made the cirrus more sparsely distributed in the tensor model we constituted and had significant sparse characteristics. Thus, the cirrus patch may satisfy a condition such as the following:

$$\|\mathcal{S}^{st}\|_0 \leq s \tag{5}$$

where $s$ was a small positive number. It could be determined by the size of the cirrus and the frequency of the cirrus appearing in the spatial-temporal tensor.

Noise patch tensor $\mathcal{N}^{st}$: The random noise in the infrared video was usually Gaussian white noise. According to (3), we could obtain that:

$$\|\mathcal{D}^{st} - \mathcal{A}^{st} - \mathcal{S}^{st}\|_F \leq n \tag{6}$$

Removing noise, the low-rank component and sparse component could be separated by solving the following optimization problem:

$$\min_{\mathcal{A}^{st}, \mathcal{S}^{st}} rank(\mathcal{A}^{st}) + \lambda \|\mathcal{S}^{st}\|_0$$

$$s.t. \, \mathcal{D}^{st} = \mathcal{A}^{st} + \mathcal{S}^{st} \tag{7}$$

According to the STT model, we could analyze the behavior of the cirrus patch image and background patch image. However, in the cirrus image, due to the different shapes of the cirrus, some shapes of the cirrus, which were similar to the background in the divided image patches, would show low rank. This would cause difficulty in distinguishing the cirrus. It was necessary to process the tensor model so that the cirrus component and the background component could be better distinguished. Therefore, we would use visual saliency to enhance the image.

*2.2. Visual Saliency Mask*

According to (7), $\lambda$ was an important regularization parameter, which will maintain equilibrium between the sparse component and low rank component. The variable $\lambda$ could be expressed as $\lambda = L/\sqrt{\min(n_1, n_2) \times n_3}$, where $n_1 \times n_2 \times n_3$ is the dimension of a tensor. It could separate the components of the cirrus and background to some extent. However, the shape of the cirrus was different from small targets. The cirrus had a large volume and different shapes. This made it difficult to select an appropriate $\lambda$.

When $\lambda$ got a smaller value, the convergence rate of the tensor model would be very low, and it would take multiple iterations to obtain the result, which was slow. Secondly, many components belonging to the background would also be decomposed into sparse components, so that the low-rank components obtained in the background were too smooth. If we increased $\lambda$ to generate a low-rank component with a higher rank, some cirrus would also enter the low-rank component, resulting in missed detection. Through experiments, it was found that for a single $\lambda$, it was impossible to determine a specific value so that the background and cirrus could be well separated.

Inspired by the patch sparse RPCA for salient motion detection in video [49], pixels could be differentiated according to whether they were cirrus pixels, and different regularization parameters were adopted for different regions. Through the pre-processing, we analyzed the image to find the possible areas of the cirrus and took different regularization parameters for the possible areas of the clouds and the cloud-free areas. For the possible areas that the cirrus may exit, a smaller regularization parameter was adopted to ensure that the sparse component did not enter the background, while for the background area, a larger regularization parameter was adopted so that the low-rank component would not enter the cirrus area. The modified TRPCA model for low-rank tensors and sparse tensors was as follows:

$$\min_{\mathcal{A}^{st}, \mathcal{S}^{st}} rank(\mathcal{A}^{st}) + \lambda \|\mathcal{S}_\Omega^{st}\|_1 + \beta \|\mathcal{S}_{\Omega^-}{}^{st}\|_1$$

$$s.t. \, \mathcal{D}^{st} = \mathcal{A}^{st} + \mathcal{S}^{st}, \mathcal{S}^{st} = \mathcal{S}_\Omega^{st} + \mathcal{S}_{\Omega^-}{}^{st} \tag{8}$$

where $\mathcal{S}_\Omega^{st}$ was the detected region with cirrus and $\mathcal{S}_{\Omega^-}{}^{st}$ was the detected region without cirrus.

Due to the use of infrared sequence images and tensors for experiments, the amount of data to be processed in the experiment was large, and the use of additional RPCA would greatly increase the operation time. Therefore, the use of visual saliency to process the cirrus area was considered.

Frequency-tuned (FT) saliency was first proposed by Achanta et al. [50], which was simple and efficient. The steps to solve the frequency modulation restriction were as follows: First, Gaussian filtering was performed on the image to remove noise and texture

details. After that, the $l_2 - norm$ of the difference between the processed image and its image mean was calculated as the final saliency map. It was defined as:

$$S = \|I_G(x,y) - I_\mu\|_2 \tag{9}$$

where $S$ is the frequency modulated saliency map, $I_G$ is the Gaussian filtered image, and $I_\mu$ is the image mean. Through the frequency-tuned saliency feature, we obtained the possible area of the cirrus, and after that we performed threshold segmentation and morphological processing on the region to obtain the mask of the multi-cloud region we needed, as shown in Figure 5. Other visual saliency methods such as spectral residual or phase of Fourier transform cannot achieve good detection results when the gray level of the cirrus in the image is low and the background has a high brightness area. Considering the complexity of the experimental scene, other saliency methods cannot meet the complexity of the experiment. Combined with the STT model proposed above, the Lap-NRSSTT model we needed was obtained. Next, we discussed the NRS used to describe the low-rank of the background.

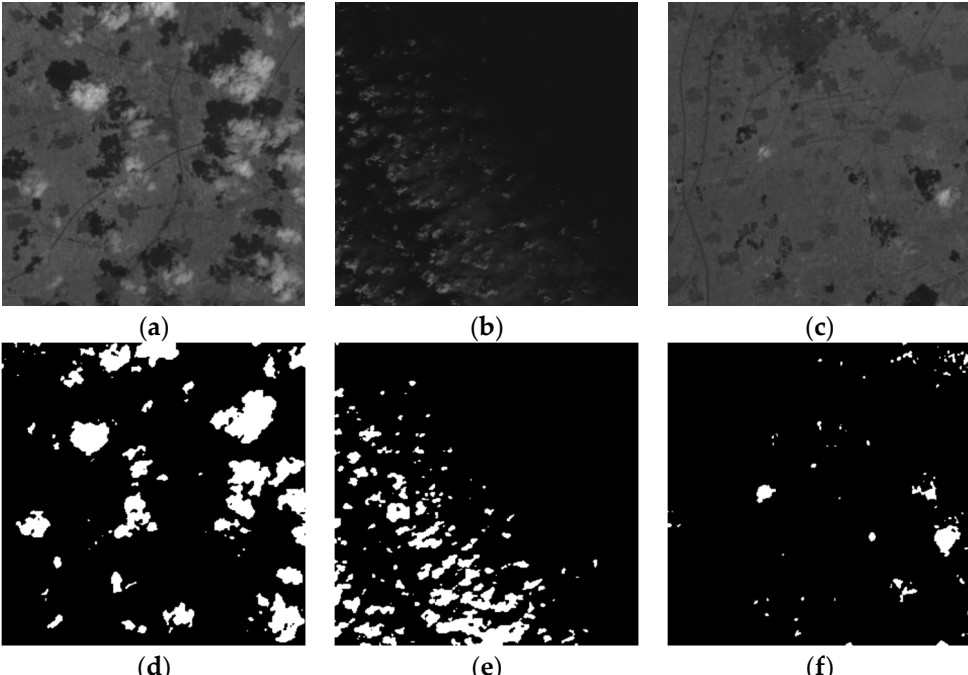

**Figure 5.** Cirrus image and corresponding mask image. (**a**–**c**) represents three classical clouds. (**d**–**f**) are the corresponding masks after morphological processing.

*2.3. Non-Convex Surrogate of Tensor Rank*

Since the rank function and $l_0 - norm$ were non-convex, it was still NP-hard to solve (7) directly. The $l_1 - norm$ was often used to replace the $l_0 - norm$, but the approximation of tensor rank was a considerable difficulty. Figuring out how to deal with the rank was the key to solve the tensor recovery problem. The TNN was a convex relaxation-of-tensor rank, which had been widely used in medical image processing, tensor completion, video denoising, etc. However, it assigned the same weight to each singular value. In infrared images, different singular values correspond to different image information. Large singular values correspond to low-frequency components, and small singular values correspond to high-frequency components. In low-rank sparse tensor decomposition, we expected that the image information corresponding to large singular values was less penalized, so the TNN could not achieve better results. Many improved methods need prior knowledge to estimate some parameters. If the estimation was not correct, it was difficult to obtain

better results. Considering the above problems, this paper used a non-convex tensor rank surrogate based on the Laplace function (Lap-NRS) to represent the rank of the tensor.

Here, Lap-NRS was defined as:

$$\|\mathcal{X}\|_\varepsilon = \sum_{k=1}^{n_3} \sum_{i=1}^{\min(n_1,n_2)} \varphi\left(\sigma_i\left(\overline{\mathcal{X}}^{(k)}\right)\right) \tag{10}$$

where $n_1 \times n_2 \times n_3$ is the dimension of tensor $\mathcal{X}$ and $\varphi(x) = 1 - e^{-x/\varepsilon}$ represents a Laplace function. $\overline{\mathcal{X}}^{(k)}$ represents the Fourier transform result in the third dimension of the $k$-th frontal slice of tensor $\mathcal{X}$ and $\sigma_i$ represents the $i$-th singular value of the corresponding slice. $\varepsilon$ represents a positive constant.

For the solution of Lap-NRS, we should consider a sub-problem as below:

$$\underset{\mathcal{B}}{\arg\min}\|\mathcal{B}\|_\varepsilon + \frac{\gamma}{2}\|\mathcal{B} - \mathcal{X}\|_F^2 \tag{11}$$

where $\mathcal{X} \in R^{n_1 \times n_2 \times n_3}$, $\gamma$ was a positive constant and the t-SVD of $\mathcal{X}$ was $\mathcal{X} = \mathcal{U} * \mathcal{S} * \mathcal{V}^T$. (11) can be solved with (12) and an extended weight singular value thresholding (WSVT) operator [51], as shown in (12) and (13):

$$\overline{\mathcal{D}}_{\frac{\nabla\varphi}{\alpha}}^{(k)} = (\overline{\mathcal{S}}^{(k)} - \frac{\nabla\varphi\left(\sigma_i\left(\overline{\mathcal{B}}^{(k)}\right)\right)}{\alpha})_+ \tag{12}$$

$$\overline{\mathcal{B}}^{(k)} = \overline{\mathcal{U}}^{(k)} * \overline{\mathcal{D}}_{\frac{\nabla\varphi}{\alpha}}^{(k)} * \overline{\mathcal{V}}^{(k)H} \tag{13}$$

where $\alpha = \frac{\gamma}{n_3}$ and $\nabla\varphi\left(\sigma_i\left(\overline{\mathcal{B}}^{(k)}\right)\right)$ represents the derivative of the Laplace function at the singular value of the $i$-th positive slice in the Fourier domain of tensor $\mathcal{B}$. Algorithm 1 describes each iteration solution in (11).

---

**Algorithm 1** Specific steps of the ADMM framework

---

Input: $\mathcal{X}$, $\mathcal{B}$;
Process:
1: Computer $\overline{\mathcal{X}} = fft(\mathcal{X}, [], 3)$;
2: for $k = 1 \ldots, [(n_3 + 1)/2]$ do
$$\left[\overline{\mathcal{U}}^{(k)}, \overline{\mathcal{S}}^{(k)}, \overline{\mathcal{V}}^{(k)}\right] = SVD(\overline{\mathcal{X}}^{(k)});$$
3: Update $\overline{\mathcal{D}}_{\frac{\nabla\varphi}{\alpha}}^{(k)}$ via (12);
4: Update $\overline{\mathcal{B}}^{(k)}$ via (13);
5: end for;
6: for $k = [(n_3 + 1)/2] + 1, \ldots, n_3$ do
$$\overline{\mathcal{B}}^{(k)} = conj(\overline{\mathcal{B}})^{n_3-k+2};$$
7: end for;
8: Compute $\overline{\mathcal{B}}^{(k+1)} = ifft(\overline{\mathcal{B}}^{(k)}, [], 3)$.

---

### 2.4. Solution of Lap-NRSSTT Mode

We used the alternating direction method of multipliers (ADMM) framework to solve this optimization problem, and the corresponding augmented Lagrangian function was as follows:

$$\mathcal{L} = \|\mathcal{A}^{st}\|_\varepsilon + \lambda\|\mathcal{S}_\Omega{}^{st}\|_1 + \beta\|\mathcal{S}_{\Omega^-}{}^{st}\|_1 + \langle Y, \mathcal{A}^{st} + \mathcal{S}^{st} - \mathcal{D}^{st}\rangle + \frac{\mu}{2}\|\mathcal{A}^{st} + \mathcal{S}^{st} - \mathcal{D}^{st}\|_F^2 \tag{14}$$

where $Y$ and $\mu > 0$ denoted the Lagrange multiplier and penalty factor, and $\lambda$ and $\beta$ were regularization parameters, which were used to coordinate the balance of each component.

ADMM decomposed the minimization problem $\mathcal{L}$ into several sub-problems, and the required $\mathcal{A}^{st}$ and $\mathcal{S}^{st}$ could be obtained through continuous iteration and updating. By solving the following subproblems, $\mathcal{A}^{st}$ and $\mathcal{S}^{st}$ were updated as follows:

$$\left(\mathcal{A}^{st}\right)^{k+1} = \mathrm{argmin}\|\mathcal{A}^{st}\|_{\varepsilon} + \frac{\mu^k}{2}\|\mathcal{A}^{st} + \left(\mathcal{S}^{st}\right)^k - \mathcal{D}^{st} + \frac{Y^k}{\mu^k}\|_F^2 \tag{15}$$

$$\left(\mathcal{S}_{\Omega}^{st}\right)^{k+1} = \mathrm{argmin}\lambda\|\mathcal{S}_{\Omega}^{st}\|_1 + \frac{\mu^k}{2}\|\left(\mathcal{A}^{st}\right)^{k+1} + \mathcal{S}_{\Omega}^{st} - \mathcal{D}^{st} + \frac{Y^k}{\mu^k}\|_F^2 \tag{16}$$

$$\left(\mathcal{S}_{\Omega^-}^{st}\right)^{k+1} = \mathrm{argmin}\beta\|\mathcal{S}_{\Omega^-}^{st}\|_1 + \frac{\mu^k}{2}\|\left(\mathcal{A}^{st}\right)^{k+1} + \mathcal{S}_{\Omega^-}^{st} - \mathcal{D}^{st} + \frac{Y^k}{\mu^k}\|_F^2 \tag{17}$$

$$\left(\mathcal{S}^{st}\right)^{k+1} = \left(\mathcal{S}_{\Omega}^{st}\right)^{k+1} + \left(\mathcal{S}_{\Omega^-}^{st}\right)^{k+1} \tag{18}$$

By introducing the relevant parameters, (15) could be solved by Algorithm 1. After updating $\mathcal{A}^{st}$, $\mathcal{S}^{st}$ could be solved by combining (16), (17), (18) and the soft threshold operator. The corresponding results are as follows:

$$\left(\mathcal{S}^{st}\right)^{k+1} = \left(S_{\Omega\lambda\mu^{-1}} + S_{\Omega^-\beta\mu^{-1}}\right)\left(\mathcal{D}^{st} - \left(\mathcal{A}^{st}\right)^{k+1} - \frac{Y^k}{\mu^k}\right) \tag{19}$$

where $S_{\Omega}(.)$ is the soft threshold operator for cloudy regions:

$$S_{\Omega\varepsilon}(x) = sign(x) \times \max(|x| - \varepsilon, 0) \tag{20}$$

The updates of $Y$ and $\mu$ are as follows:

$$Y^{k+1} = Y^k + \mu^k\left(\left(\mathcal{A}^{st}\right)^{k+1} + \left(\mathcal{S}^{st}\right)^{k+1} - \mathcal{D}^{st}\right) \tag{21}$$

$$\mu^{k+1} = \rho\mu^k \tag{22}$$

Finally, the specific process of solving the model by ADMM was shown in Algorithm 2.

---

**Algorithm 2** ADMM for solving the proposed model.

---

Input: $\mathcal{D}^{st}$, $\lambda$, $\beta$;
Initialize: $\left(\mathcal{S}^{st}\right)^0 = \left(\mathcal{A}^{st}\right)^0 = Y^0 = 0$, $\mu^0 = 0.00020$, $\rho = 1.05$, $k = 0$ $tol = 1e^{-7}$;
While not converge do
1: Update $\left(\mathcal{A}^{st}\right)^{k+1}$ by Algorithm 1;
2: Update $\left(\mathcal{S}^{st}\right)^{k+1}$ by (19);
3: Update $Y^{k+1}$ by (21);
4: Update $\mu^{k+1}$ by (22);
5: Check the convergence conditions
$\frac{\|\left(\mathcal{A}^{st}\right)^{k+1} + \left(\mathcal{S}^{st}\right)^{k+1} - \mathcal{D}^{st}\|_F}{\|\mathcal{D}^{st}\|_F} < tol$ or $\|\left(\mathcal{S}^{st}\right)^k\|_0 = \|\left(\mathcal{S}^{st}\right)^{k+1}\|_0$;
6: Update k: $k = k + 1$;
7: Output: $\left(\mathcal{A}^{st}\right)^k$, $\left(\mathcal{S}^{st}\right)^k$.

---

### 2.5. Steps of the Method

Figure 6 showed the entire course of infrared cirrus detection scheme based on proposed measure, which was described as follows:

1.  Inputting the image. Given the current frame $f_i \in R^{n_1 \times n_2}$ and its adjacent $2 \times s$ frames $f_{i+y} \in R^{n_1 \times n_2} (y \in (-s, 1) \cup (1, s))$. Each frame of the image traversed the whole image through a sliding window of size $m \times m$ to obtain an $Num$ image patch;

2. Construction of the STT model. For each image of $f_i \in R^{n_1 \times n_2}$, an STT $\mathcal{D}^{st} \in R^{m \times m \times h}$ was constructed according to the proposed model, where $h = 9 \times (2 \times s + 1)$;

3. Using visual saliency to separate cloudy areas and cloudless areas. For each frontal slice of the input STT $\mathcal{D}^{st}$, the visual saliency was calculated respectively, and STT $\mathcal{S}_\Omega{}^{st} \in R^{m \times m \times h}$ and $\mathcal{S}_{\Omega^-}{}^{st} \in R^{m \times m \times h}$ containing the prior information of the cirrus is obtained;

4. Separate background and cirrus. Taking STT $\mathcal{D}^{st}$ as the input tensor, $\mathcal{D}^{st}$ was decomposed into background patch tensor $\mathcal{A}^{st}$ and cirrus patch tensor $\mathcal{S}^{st}$ by Algorithm 2;

5. Image reconstruction and cirrus detection. The obtained background patch tensor $\mathcal{A}^{st}$ and cirrus patch tensor $\mathcal{S}^{st}$ were reconstructed to obtain the background image $f_a \in R^{n_1 \times n_2}$ and the cirrus image $f_s \in R^{n_1 \times n_2}$. Then the detection result was obtained by one or more set threshold segmentation.

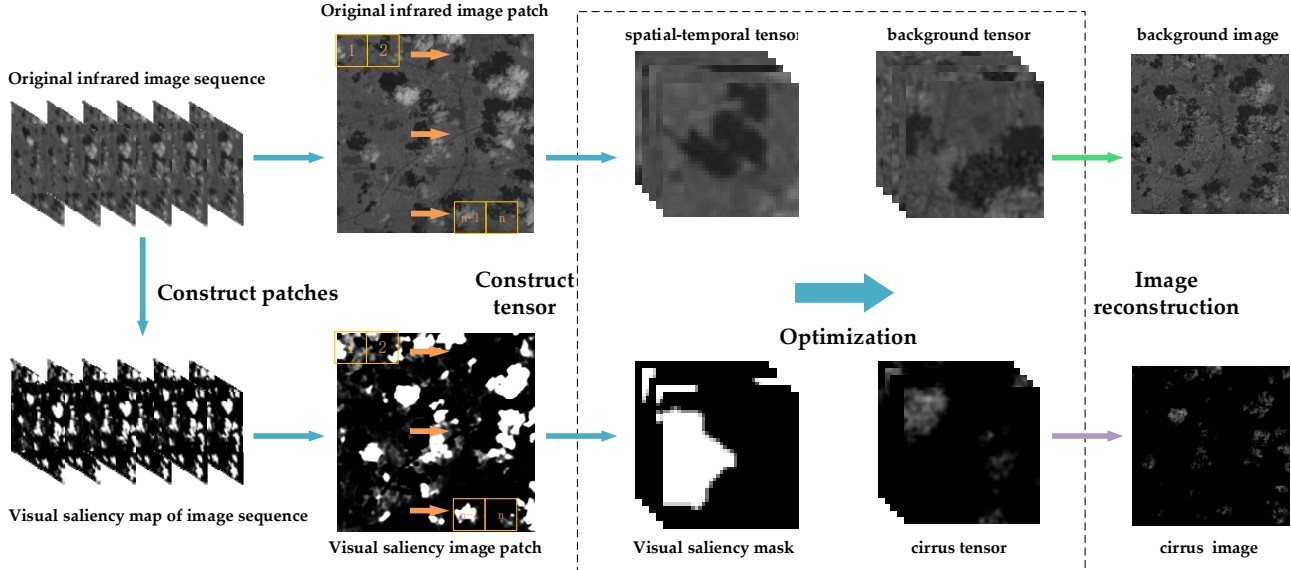

**Figure 6.** The whole process of the proposed algorithm.

## 3. Experimental Results and Analysis

### 3.1. Experimental Preparations

This paper tests six representative cirrus infrared image sequences, as shown in Figure 7.

The experimental data are derived from the near-infrared band of the Landsat8 dataset [52]. From the diagram, Sequence a included some cirrus with a large imaging area and dense distribution. Sequence b is dotted, densely distributed in the upper right, and sparsely distributed in the lower left. Sequence c is a sparsely distributed massive cirrus image. Sequence d is a punctate cirrus concentrated in the lower right. Sequence e is a cirrus showing a silky state. Sequence f is a cirrus image with dense distribution of the whole image. These six sequences contain the form and distribution of most cirrus, making the experimental results more universal.

To receive appraisal for the proposed method objectively, it should be compared with the false alarm source detection method based on other optimization-based methods. Objective evaluation methods include receiver operating characteristic (ROC) curve, Precision-Recall (PR) curve, and F-measure.

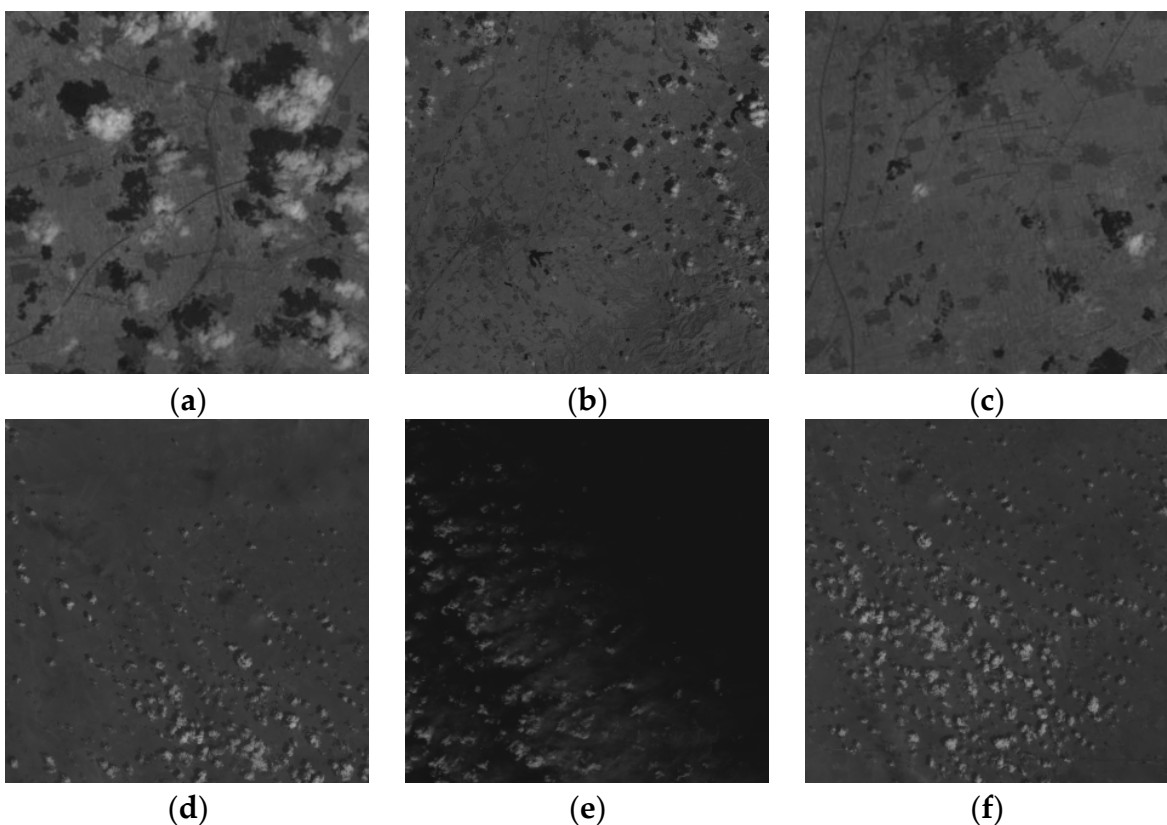

**Figure 7.** (**a**–**f**) represent 6 original infrared images of different scenes.

### 3.2. Evaluation Metrics

The ROC curve and the PR curve are supervised evaluations, and real cirrus images need to be manually marked. The corresponding concepts of TP, FP, FN and TN are illustrated by the Table 1.

**Table 1.** Concepts of TP, FP, FN and TN.

|  | **Actual Positive** | **Actual Negative** |
|---|---|---|
| predicted positive | TP | FP |
| predicted negative | FN | TN |

TP represents the number of pixels that are considered to be cirrus in the detection results and marked as cirrus in the ground truth. FP represents the number of pixels that are considered to not be cirrus in the detection results but marked as cirrus in the ground truth. FN represents the number of pixels that are considered to not be cirrus in the detection results but marked as cirrus in the ground truth. TN represents the number of pixels that are considered to not be cirrus in the detection results and not marked as cirrus in the ground truth.

The above four indicators cannot accurately represent the performance of the detection method. A detection measure with a large TP value may also have large FN and FP values. Therefore, it is necessary to consider combining multiple indicators to evaluate the performance of detection. TPR and FPR are two commonly used evaluation indexes, and the corresponding definitions are as below:

$$\text{TPR} = \frac{\text{TP}}{\text{TP} + \text{FN}} \tag{23}$$

$$FPR = \frac{FP}{FP + TN} \tag{24}$$

The ROC curve sets the abscissa as FPR and the ordinate as TPR. When a ROC curve obtained by a certain detection measure is close to the upper left corner, the area under the curve (AUC) is large, indicating that the measure has excellent detection performance. However, if the detection result image is fully marked as the cirrus, the TPR and FPR will be set to 1, so the evaluation effect of the ROC image is still good. In this case, we need another curve to evaluate the detection measure.

PR curves set the abscissa as recall and ordinate as precision. The corresponding definitions are as below:

$$precision = \frac{TP}{TP + FP} \tag{25}$$

$$recall = \frac{TP}{TP + FN} \tag{26}$$

When the PR curve is close to the upper right corner, the area under the curve is large, indicating that the method has excellent detection performance.

Although there is no necessary relationship between recall and precision, the two indicators are mutually restrictive. A useful measure to evaluate the conflict between them is F-measure (also known as F-score). F-measure is the weighted harmonic mean of precision and recall and can be described as follows:

$$F - measure = \frac{(\alpha^2 + 1)(precision \times recall)}{\alpha^2 \times precision + recall} \tag{27}$$

when $\alpha^2$ is set to 1, the F-measure is also called F1-score.

### 3.3. Parameter Analysis

The proposed model contains some key parameters, such as image patch size, and regularization parameters $\lambda$ and $\beta$. These parameters will affect the detection performance of the proposed model for different forms of cirrus and the robustness to various scenarios. Therefore, To achieve better performance in different scenarios, appropriate parameter settings should be selected. We selected the appropriate parameter by single variable measures and comparative experiments, and then analyzed the result according to the ROC curve and PR curve. In the experiment, when studying the optimization of a certain parameter, other parameters are fixed, and one of the parameters is adjusted.

### 3.3.1. Patch Size

The sliding window size represents the positive slice size of the STT, so the image patch size not only has a significant influence on the accuracy of the cirrus detection, but also affects the calculation time. On the one hand, we want to use larger image patches to ensure that the cirrus is sparse enough in a forward slice, but larger image patches will increase the computational complexity of singular value decomposition, thus increasing the computational time. On the other hand, a smaller image patch will reduce the processing complexity of the model, but it will also reduce the sparsity of the volume cloud, resulting in the need for more iterations to reach the iterative stopping condition in the process of optimization. To study the influence of image patch size, we changed the patch size from 40 to 80. The ROC curve and PR curve of the 6 test sequences are shown in Figures 8 and 9. Considering the ROC curve, PR curve and running time, the size of the image patch is set to 60.

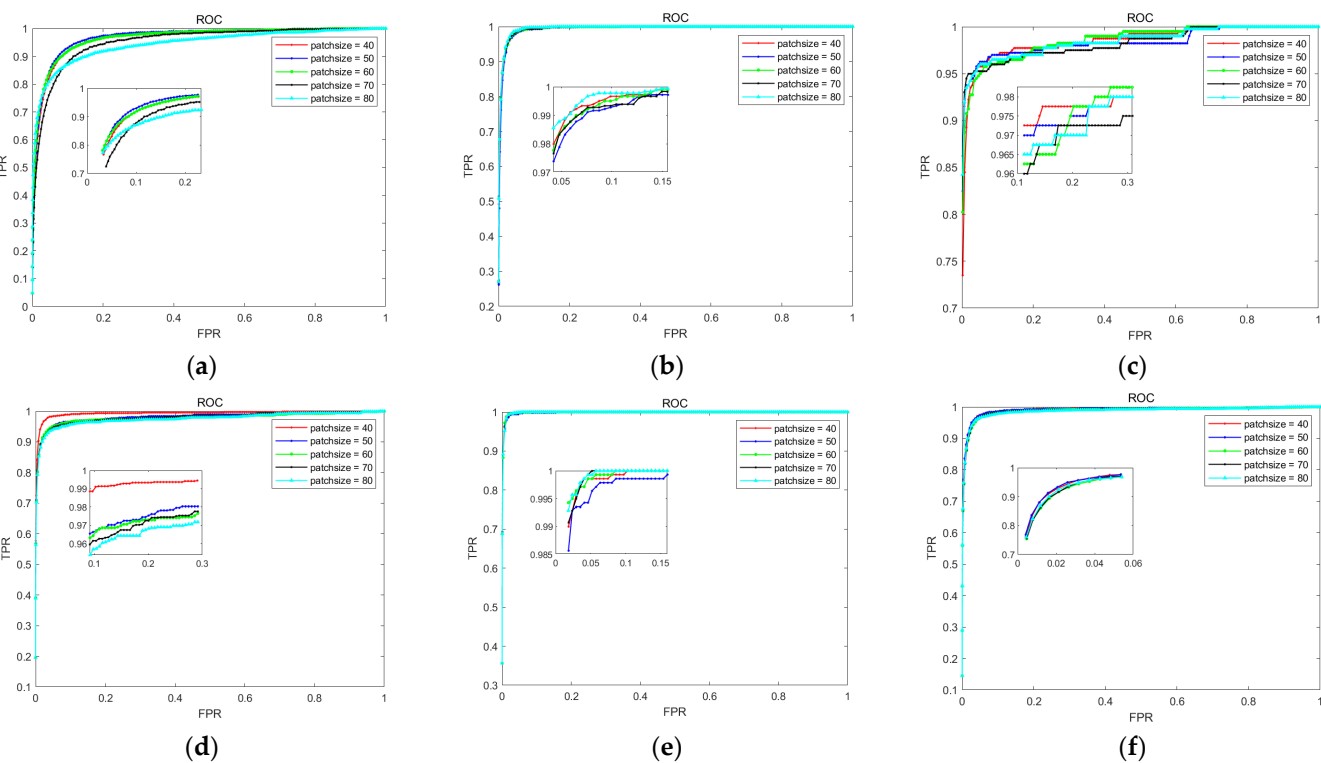

**Figure 8.** ROC curves of different patch sizes in the 6 sequence images. (**a**) Sequence 1; (**b**) Sequence 2; (**c**) Sequence 3; (**d**) Sequence 4; (**e**) Sequence 5; (**f**) Sequence 6.

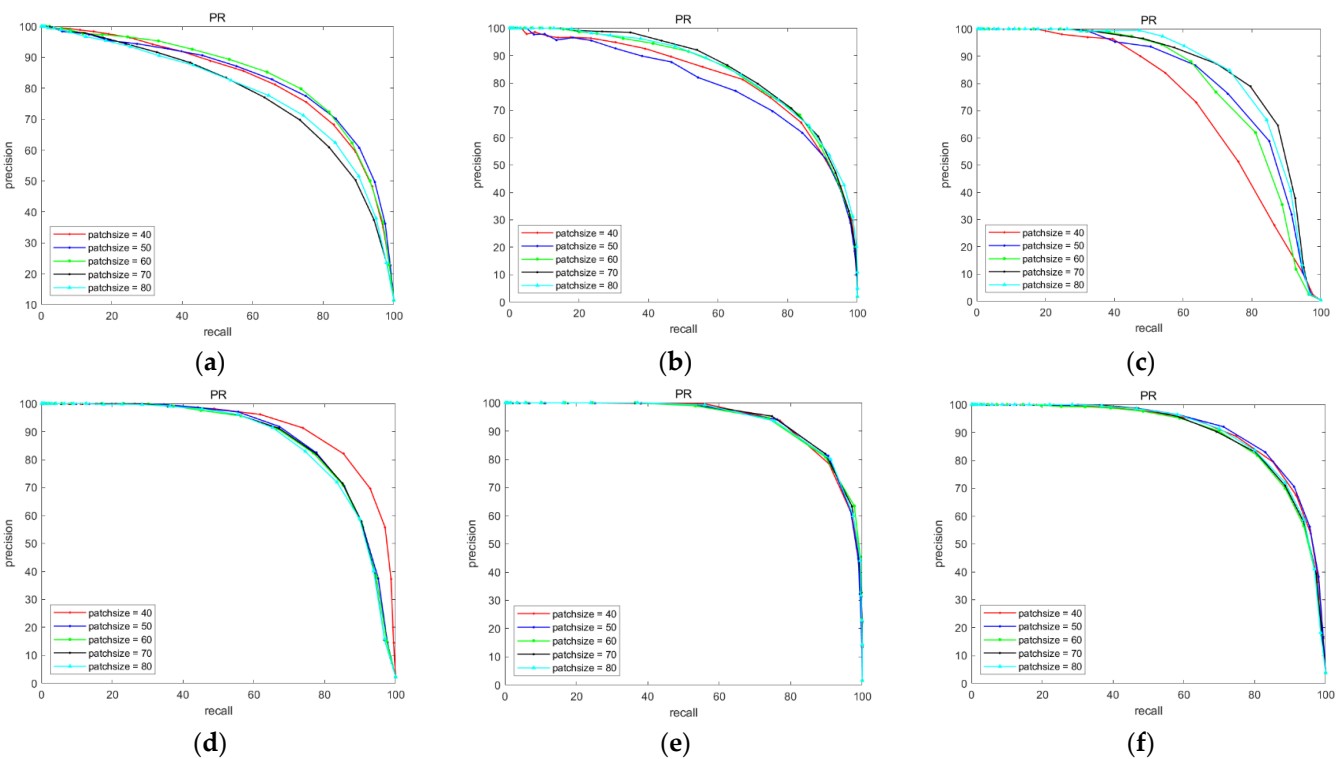

**Figure 9.** PR curves of different patch sizes in the 6-sequence images. (**a**) Sequence 1; (**b**) Sequence 2; (**c**) Sequence 3; (**d**) Sequence 4; (**e**) Sequence 5; (**f**) Sequence 6.

#### 3.3.2. Regularization Parameter

In the proposed model, the regularization parameters $\lambda$ and $\beta$ control the balance between low-rank components and sparse components as the weights of cloudy and cloud-free regions. According to the empirical value, We set $\beta = k \times \lambda$. A larger k means there are more small clouds in the cloud-free area. However, the mask in our study contains mostly cloudy areas, and only a few clouds may be in the cloud-free area, so the k is set to 25.

For the regularization parameter $\lambda$ in the cloudy region, similar to the proposed tensor model for infrared detection, $\lambda$ is set to $\lambda = L / \sqrt{\min(n_1, n_2) \times n_3}$. Change $L$ from 0.02 to 0.1. The ROC curve and PR curve of the 6 test sequences are shown in Figures 10 and 11. For a large-volume cirrus, changing the regularization parameter has little effect on the detection ability, but the corresponding operation time will increase due to the increase in optimization times. For a smaller cirrus, when $\lambda$ continues to increase, the components belonging to the cloud will be separated from the low-rank components, resulting in a decrease in detection performance. Considering the ROC curve, PR curve and running time, when $\lambda = 0.02$, the best performances can be obtained.

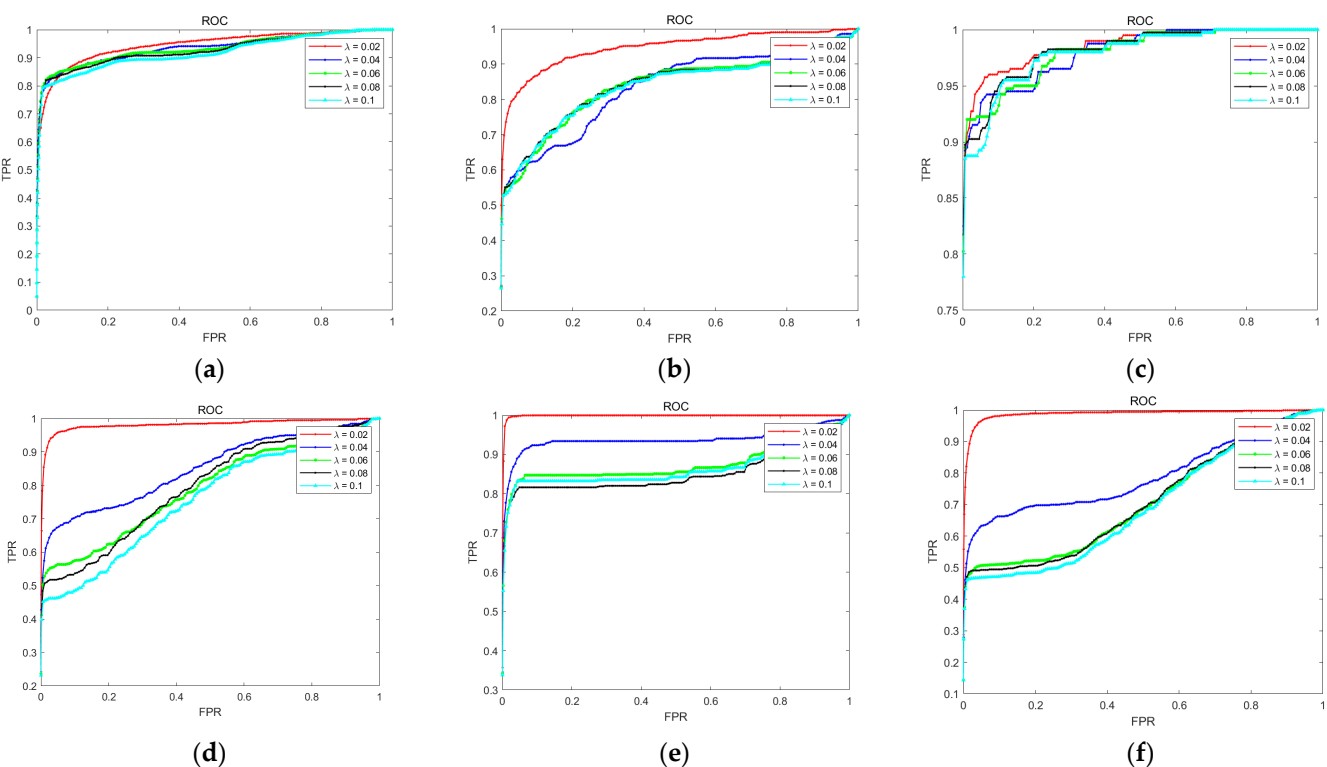

**Figure 10.** ROC curves of different regularization parameters in the 6 sequence images. (**a**) Sequence 1; (**b**) Sequence 2; (**c**) Sequence 3; (**d**) Sequence 4; (**e**) Sequence 5; (**f**) Sequence 6.

#### 3.4. Method Comparison

It is difficult to qualitatively analyze the results of cirrus detection with different methods. (Some methods may perform better in local cirrus detection and perform poorly in other locations of the image. In this case, qualitative analysis is difficult to compare the advantages and disadvantages of various methods). Therefore, in this section, the quantitative analysis method will be used to compare the methods. In the proposed method, patch size is set to 60, regularization parameter $\lambda = 0.02$, and we use 5 frames (2 frames before the current frame, the current frame, and 2 frames after the current frame) to build a tensor. The comparison indicators are the ROC curves, PR curves and F-measure values of various methods. The following methods are compared: IPI [44], LOGTFNN [30], PSTNN [6], TMESNN [29], KSVD fractal [28] and DivisorstepTP [53].

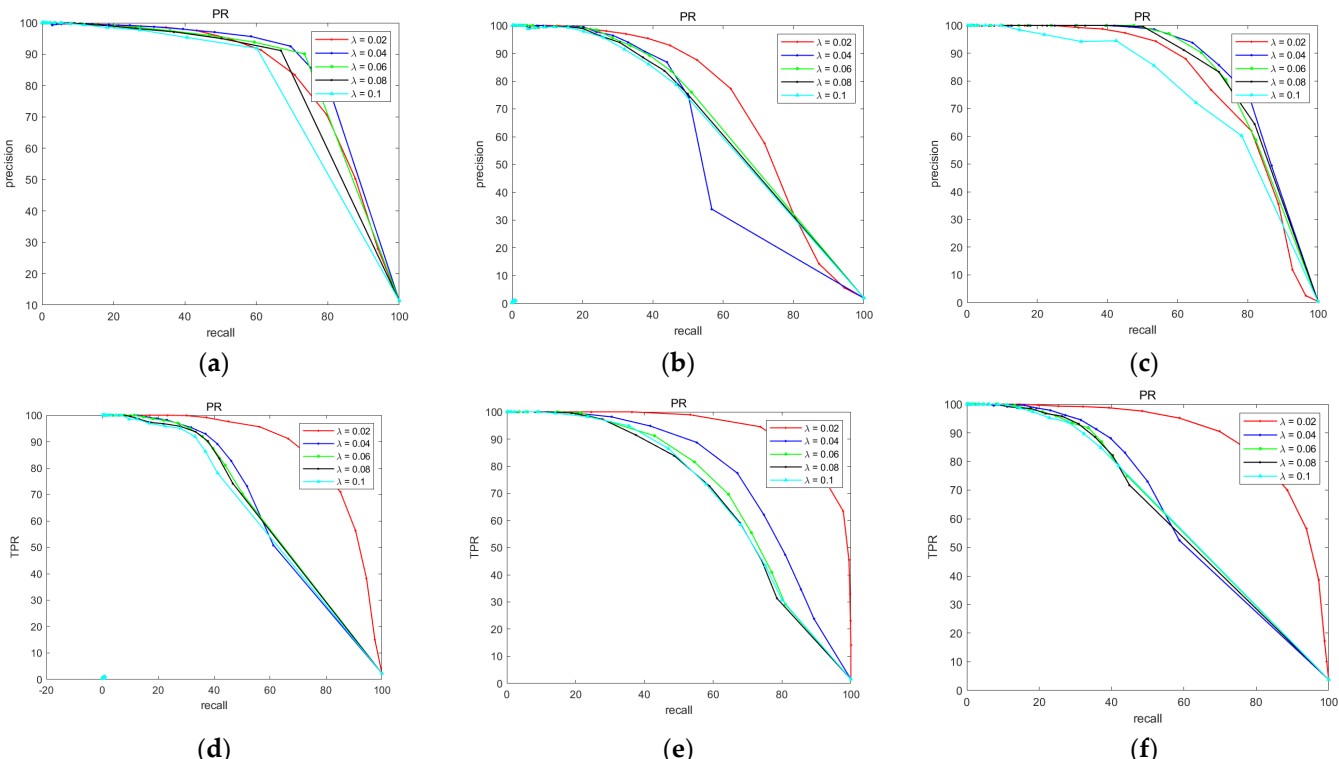

**Figure 11.** PR curves of different regularization parameters in the 6-sequence images. (**a**) Sequence 1; (**b**) Sequence 2; (**c**) Sequence 3; (**d**) Sequence 4; (**e**) Sequence 5; (**f**) Sequence 6.

Figures 12 and 13 shows the ROC curve and PR curve of all methods. As discussed above, the traditional optimization-based method uses a single regularization parameter, and the detection effect is poor for a larger cirrus. The method proposed in this paper can not only detect a large cirrus, but also has better detection performance than other methods based on optimization for a small cirrus with different distributions.

Table 2 shows the F-measure the above methods in 6 test image sequences. The bold values indicate the maximum value. It can be seen that the proposed method maintains a good detection effect.

**Table 2.** F-measure of six methods.

| . | IPI | LOGTFNN | PSTNN | KSVD Fractal | TMESNN | DivisorstepTP | Proposed |
|---|---|---|---|---|---|---|---|
| Seq1 | 0.1257 | 0.3324 | 0.0059 | 0.6303 | 0.1430 | 0.2986 | 0.8374 |
| Seq2 | 0.7008 | 0.4495 | 0.5711 | 0.7478 | 0.6832 | 0.3420 | 0.7612 |
| Seq3 | 0.2786 | 0.3706 | 0.2554 | 0.5563 | 0.1777 | 0.1387 | 0.8129 |
| Seq4 | 0.7645 | 0.6785 | 0.7645 | 0.7960 | 0.6740 | 0.5842 | 0.8443 |
| Seq5 | 0.6863 | 0.5587 | 0.3942 | 0.7548 | 0.6129 | 0.5405 | 0.8874 |
| Seq6 | 0.6303 | 0.6769 | 0.7395 | 0.8205 | 0.6463 | 0.5849 | 0.8488 |

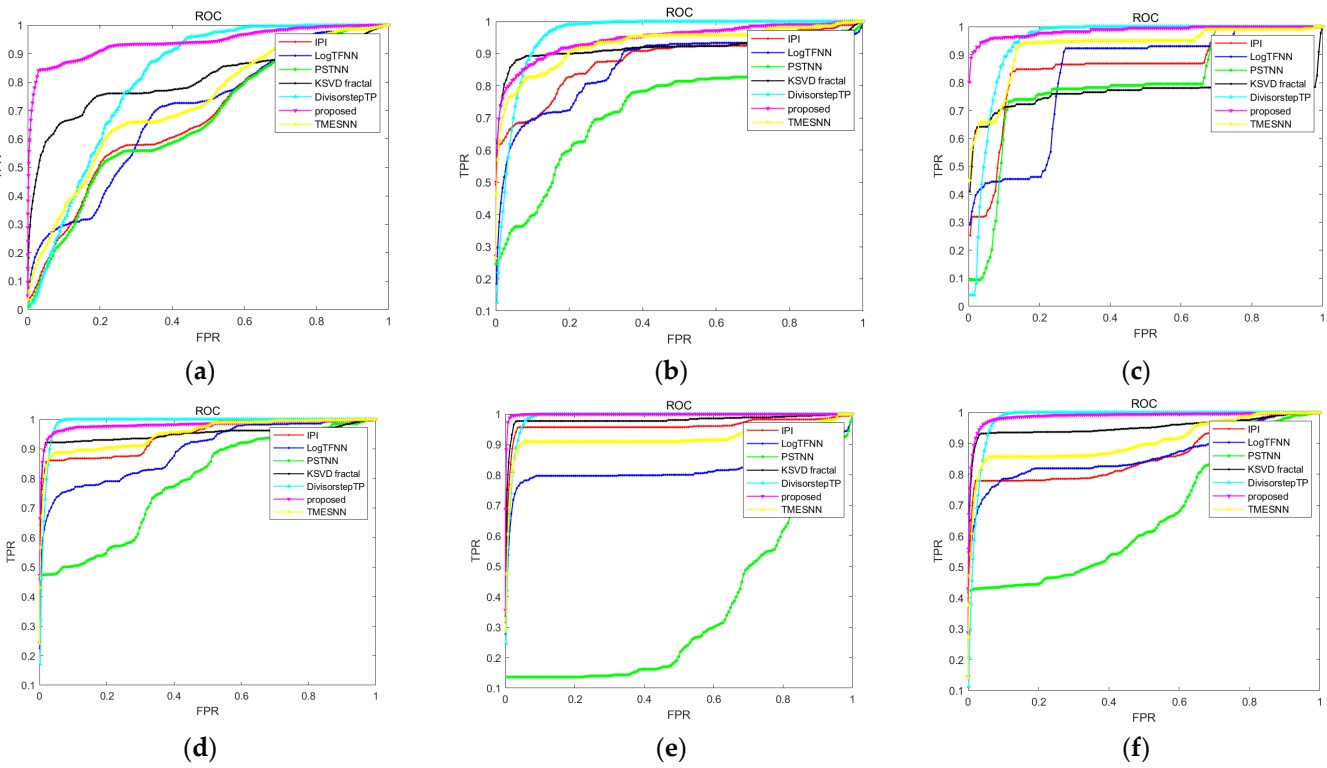

**Figure 12.** ROC curves of different regularization parameters in the 6-sequence images. (**a**) Sequence 1; (**b**) Sequence 2; (**c**) Sequence 3; (**d**) Sequence 4; (**e**) Sequence 5; (**f**) Sequence 6.

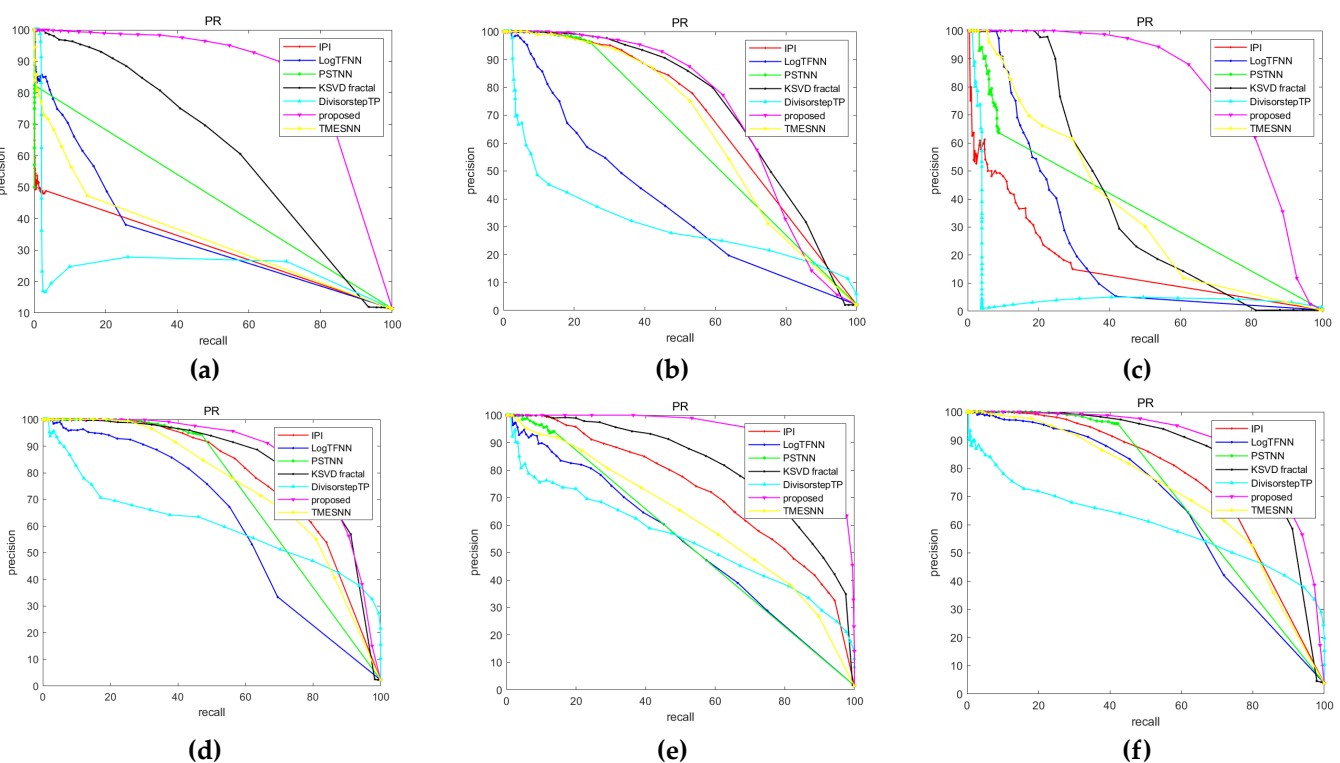

**Figure 13.** PR curves of different methods in the 6-sequence images. (**a**) Sequence 1; (**b**) Sequence 2; (**c**) Sequence 3; (**d**) Sequence 4; (**e**) Sequence 5; (**f**) Sequence 6.

## 4. Discussion

For cirrus detection, researchers have proposed methods based on physical model classes, machine learning and low-rank sparse decomposition. However, the use of physical model methods requires atmospheric radiation, geophysics and other related knowledge, while machine learning methods require a certain amount of data and corresponding labels as training samples, which is costly. The low-rank sparse decomposition measure fully utilizes the attributes of background and cirrus, which is closer to the essence of the image. In particular, the derived TRPCA model transferred the infrared image into a third-order tensor, and then used the low-rank nature of the infrared background and sparsity of the cirrus for detection, which results in a great detection accuracy.

The traditional IPT model is used to detect small targets. Compared with small targets, cirrus are larger and have a certain shape. Therefore, it is not feasible to directly apply the IPT model to cirrus detection. To ensure that the infrared tensor model can be successfully used for cirrus detection, we propose a cirrus detection method based on non-convex tensor rank surrogate. Firstly, to enhance the low rank of the cirrus tensor, time information is introduced to construct an STT model, which enhances the sparsity of the cirrus in a certain range. Secondly, after the experiment, it is found that a single balance coefficient cannot achieve the optimization solution. After that, the visual saliency is introduced as a priori to divide the cloudy and cloudless regions, and different balance coefficients are given. To solve the measure, an optimization method based on ADMM is designed. Finally, the detection results are segmented by one or more adaptive thresholds to obtain the final detection results.

This method is quantitatively compared with IPI, LOGTNN, TMESNN, PSTNN, KSVD fractal and DivisorstepTP. In Figures 12 and 13 and Table 2, the ROC curve for the proposed method is closer to the upper left corner, and the PR curve of this method is closer to the upper right corner. In Table 2, the F-measure shows that the proposed method can achieve the best performance in all test sequences. In summary, the NTS measure has a great accuracy for cirrus sources.

## 5. Conclusions

We proposed an NTS-based cirrus detection measure, which focused on the accurate representation of background rank of infrared image and sparse enhancement of cirrus. For the purpose of representing the tensor rank, we used t-SVD decomposition and extended the non-convex surrogate based on Laplace function. To enhance the sparsity of the cirrus, by introducing spatial-temporal patches, an STT model conforming to the characteristics of infrared images was obtained through experimental comparison. By using the visual saliency of cirrus, a mask based on cirrus was generated, so that the improved model could be used for cirrus detection, which laid a foundation for the detection of false alarms in a similar large volume. To solve the model, an optimization method based on ADMM was designed. By combining the optimization function with ADMM, the problem was solved and its iteration was optimized. The experimental results showed that the measure could detect different forms of cirrus in different scenarios, and its quality indicators such as ROC curve, PR curve and F-measure also showed better performance than other optimization-based algorithms.

However, our proposed method is based on sequence images, and the construction of a spatial-temporal tensor is complex. In the process of solving the model, multiple singular value decomposition and optimization iterations are needed. Therefore, there is a lack of real-time performance. Additionally, our method is designed for small sample data and lacks scene applicability compared to deep learning methods. We can improve its performance by combining traditional features with deep learning in the future.

**Author Contributions:** All authors have contributed substantially to, and are in agreement with the content of, the manuscript. Conception/design, provision of study materials, and the collection and/or assembly of data: Conceptualization, S.X. and Z.P.; methodology, S.X. and F.L. and Z.P.; data analysis and interpretation: S.X. and F.L.; manuscript preparation: S.X. and Z.P.; final approval of the manuscript: S.X., F.L. and Z.P. The guarantor of the paper takes responsibility for the integrity of the work as a whole, from its inception to publication. All authors have read and agreed to the published version of the manuscript.

**Funding:** This work was supported by Natural Science Foundation of Sichuan Province of China (Grant No.2022NSFSC40574) and partially supported by National Natural Science Foundation of China (Grant No.61775030, Grant No.61571096).

**Data Availability Statement:** All data included in this study are available upon request by contact with the corresponding author.

**Acknowledgments:** The authors would thank Geospatial Data Cloud Website for providing the experimental data, and appreciate the support from the Laboratory of Imaging Detection and Intelligent Perception.

**Conflicts of Interest:** The authors declare no conflict of interest.

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
