# Peer review of "Infrared Cirrus Detection Using Non-Convex Rank Surrogates for Spatial-Temporal Tensor"

_remotesensing, doi:10.3390/rs15092334_

Round 1

Reviewer 1 Report

Specific to the problem of cirrus detection in complex scenes, the authors proposed an algorithm based on spatial-temporal tensor. However, there is still a room for improvement in the paper organization and presentation. More specifically, there are several issues that need to be addressed before a possible publication.

1. There may be some grammatical or spelling errors in the paper.

2. Some terms used in the text were inconsistent, ones of which were edited by MathType and others were not. The authors should correct them.

3. In 2.2, why choose Frequency tuned saliency to build the mask?

4. In 3.2.2, why is k set to 25? Please give some simple explanation or discuss this parameter in this paper.

5. The baseline methods used in this paper are all based on optimization, and only five baseline methods are insufficient. Therefore, the author is suggested to compare and add some other methods. Please refer to but not limited to the following literature:

[1] Lyu, Y.; Peng, L.; Pu, T.; Yang, C.; Wang, J.; Peng, Z. Cirrus Detection Based on RPCA and Fractal Dictionary Learning in In-frared imagery. Remote Sensing 2020, 12, 142.

6. Please provide more explanation on the innovative nature of the proposed algorithm. For example, how the performance of the proposed algorithm outperforms existing methods.

Author Response

We would like to thank the reviewers for their valuable advice and suggestions for this article.We have carefully considered your comments and suggestions and have made significant changes to improve the quality of the manuscript. Please see the attachment to get a point-by-point response to your comments, along with the corresponding modifications made to the manuscript.

Reviewer 2 Report

Infrared cirrus detection plays a significant role in earth observation infrared system. A measure for cirrus detection based on visual saliency and non-convex spatial-temporal tensor rank surrogate is proposed in this paper. The organization structure of the paper is reasonable and the research technology route is clearly stated. Experimental results show the effectiveness and advantages of the proposed algorithm. However, there are still some deficiencies in the paper. Please revise it carefully.

1The article talks about cirrus detection, but small target detection occurs many times in the article. The two concepts appear alternately, which is easy to confuse the readers with the real research object. It is suggested to unify the research object.

2The ROC curve in Figure 6 is difficult to distinguish clearly, so it is recommended to add an enlarged figure.

3The following five optimization-based methods are compared: IPI, LOGTFNN, PSTNN, RIPT, TMESNN. No corresponding references can be found and there is no introduction in the introduction part; The comparison method is suggested to be compared with the new method proposed recently.

4The marking format of references is not uniform.

5How about the processing time of the proposed algorithm and whether it can meet the real-time requirements? If it has advantages, please give the comparison results. If it has no advantages, it is recommended to give explanations.

Author Response

(The authors gave the same response as above.)

Reviewer 3 Report

Review Comments

[Title]

Infrared Cirrus Detection Using Non-Convex Rank Surrogates for Spatial-Temporal Tensor

[Summary]

The authors propose a tensol-based discrimination method of cirrus and non-cirrus area from infrared image.

[Broad Comment]

The manuscript is generally well-organized. Section 4 and 5 can be merged as “summary and conclusions”. As listed below, there are many careless misses in the manuscript. So, I recommend you to check the manuscript carefully in the revision.

[Specific Comments]

Line 24-25

   The words “achieve” and “better” duplicate.

Line 33

   infrared …  -> Infrared … (Begin in Capital)

Line 34

   Small size … -> small size …

Line 50

   … properties, Since … -> … properties, since …

Line 62

   … has and non-local … -> … has non-local …

Line 83

   Tensor, As a … -> Tensor, as a …

Line 153

   a sliding window … -> A sliding window …

Line 247

   Cirrus had large volume large and … -> Cirrus had large volume and … (Remove the 2nd “large”) 

Line 310

   Many improved methods needed … -> Many improved methods are needed …

Line 552

   we …  -> We … (Begin in Capital)

Author Response

(The authors gave the same response as above.)
